# Preferred Lung Cancer Screening Modalities in China: A Discrete Choice Experiment

**DOI:** 10.3390/cancers13236110

**Published:** 2021-12-03

**Authors:** Zixuan Zhao, Lingbin Du, Le Wang, Youqing Wang, Yi Yang, Hengjin Dong

**Affiliations:** 1Center for Health Policy Studies, School of Public Health, Zhejiang University School of Medicine, Hangzhou 310058, China; zxzhao@zju.edu.cn (Z.Z.); 12018568@zju.edu.cn (Y.Y.); 2Department of Cancer Prevention, Institute of Cancer and Basic Medicine, Chinese Academy of Sciences, Cancer Hospital of the University of Chinese Academy of Sciences (Zhejiang Cancer Hospital), Hangzhou 310022, China; dulb@zjcc.org.cn (L.D.); wangle@zjcc.org.cn (L.W.); wangyq@zjcc.org.cn (Y.W.); 3The Fourth Affiliated Hospital, Zhejiang University School of Medicine, Yiwu 322000, China

**Keywords:** discrete choice experiment, lung cancer, screening modality, China

## Abstract

**Simple Summary:**

Few results from public attitudes for lung cancer screening are available both in China and abroad. The aims of this study were to explore whether preferences were related to respondent characteristics and identify which kinds of respondents were more likely to opt out of any screening. Preferred screening modality in this study was inconsistent with current Chinese practice. Screening interval was the main determinant of preferred lung cancer screening modality in both the general respondents and in subgroups, this poses a considerable challenge to the implementation of a sustainable, regular screening programme. In addition, those with no endowment insurance were more likely to opt out; indicating that a promotion of financial support is needed to reduce inequalities of attendance of disadvantaged elderly.

**Abstract:**

This study aimed to identify preferred lung cancer screening modalities in a Chinese population and predict uptake rates of different modalities. A discrete choice experiment questionnaire was administered to 392 Chinese individuals aged 50–74 years who were at high risk for lung cancer. Each choice set had two lung screening options and an option to opt-out, and respondents were asked to choose the most preferred one. Both mixed logit analysis and stepwise logistic analysis were conducted to explore whether preferences were related to respondent characteristics and identify which kinds of respondents were more likely to opt out of any screening. On mixed logit analysis, attributes that were predictive of choice at 1% level of statistical significance included the screening interval, screening venue, and out-of-pocket costs. The preferred screening modality seemed to be screening by low-dose computed tomography (LDCT) + blood test once a year in a general hospital at a cost of RMB 50; this could increase the uptake rate by 0.40 compared to the baseline setting. On stepwise logistic regression, those with no endowment insurance were more likely to opt out; those who were older and housewives/househusbands, and those with a health check habit and with commercial endowment insurance were less likely to opt out from a screening programme. There was considerable variance between real risk and self-perceived risk of lung cancer among respondents. Lung cancer screening uptake can be increased by offering various screening modalities, so as to help policymakers further design the screening modality.

## 1. Introduction

Lung cancer is one of the leading causes of cancer-related deaths in China, with a 5-year survival rate of only 19.8% [1,2]. In the past 30 years, mortality due to lung cancer in China has increased by 465% according to the Third National Mortality Retrospective Sampling Survey [3]. Early detection and diagnosis of (non-small cell) lung cancer can lead to appropriate treatment for patients, which can substantially decrease both cancer-related and all-cause mortality [4]. Cancer screening is aimed at detecting asymptomatic individuals at an early stage to reduce cancer mortality, which has caused widespread concern worldwide. However, uptake rates of lung cancer screening programmes from 2013 to 2018 remained 34.41%, 37.25%, and 48.21% in urban areas of Shanxi, Henan, and Zhejiang Provinces, respectively [5,6,7]. High uptake is an essential component of a successful screening programme [8]. Therefore, it is of great significance to explore what aspects could help to increase the uptake of a lung cancer screening in order to establish a more attractive screening programme in China.

Most of the previous research in this area was conducted on imaging-based screening modalities; it was found that chest radiography provided no additional survival benefit over sputum cytology [9] and that low-dose computerised tomography (LDCT) reduced cancer-related mortality by 20% compared with chest radiography [10,11]. Blood biomarkers are screening modalities with a low radiation burden. A clinical study of 596 cases of small pulmonary nodules in China confirmed that the use of blood biomarkers for screening could significantly improve the sensitivity and specificity of early detection of lung cancer and could be combined with LDCT to assist in the diagnosis of lung cancer [12]. Radiation from an LDCT scan for lung cancer is equivalent to the normal background radiation that a person receives in a month [13]. However, as concerns about radiation risk and over-diagnosis emerged [14], the screening tools and intervals used to screen for lung cancer began to affect the acceptability of screening programs. Other potential drivers of non-uptake are out-of-pocket costs and the quality of the screening venues.

Discrete choice experiments (DCEs), one of the stated preferred methods, have been increasingly used in health economics studies to elicit patient preferences [15]. The identification of public preferences is of great importance in informing policy decision-making and improving attendance at public healthcare interventions or programmes [16]. Numerous studies have used DCEs to investigate preferences of people for various types of cancer screening, such as breast [17], colorectal [18,19,20], prostate [21,22,23], and cervical cancer [24]. Previous studies on lung cancer both in China and abroad have mainly focused on patients’ treatment preferences [25,26,27,28]. Norman et al. conducted the best-worst experiment in 521 Australians between the age of 50 and 80 years with a history of cigarette smoking to measure their preferences in lung cancer screening; they found that respondents preferred blood tests, a location that was close to home, receiving results quickly, and minimal radiation from CT scans [13]. However, no results from DCEs for lung cancer screening are available in China for now. This study aims to obtain quantitative insights into the relative importance of attributes of lung cancer screening by administering a DCE questionnaire in a Chinese population with a high risk of lung cancer, so as to help policymakers design the screening modality by selecting the attribute levels that maximise screening attendance. We wished to answer two questions: (1) What makes a test attractive to potential recipients? (2) Who would more likely to opt-out of any screening in China?

## 2. Materials and Methods

### 2.1. Study Population

A high-risk sample from the general population in Wenling City, Zhejiang Province, was enrolled for this analysis through a local lung cancer-screening programme and written informed consent was obtained once the enrolment complete. Men and women aged between 50 and 74 years with no history of lung cancer and who met the inclusion criteria for participating in a lung cancer-screening programme were included. Specifically, inclusion criteria included (1) being an ever-smoker or quitting smoking in 15 years and living with smokers for up to 20 years; (2) having a history of related diseases (e.g., emphysema, chronic obstructive pulmonary disease, etc.); (3) having experienced occupational exposure (e.g., asbestos, radon, beryllium, uranium, chromium, etc.); and (4) having a family history of cancer. Individuals were recruited if they fulfilled any one of the above criteria. Participants with fatal diseases (e.g., severe cardiovascular and cerebrovascular diseases, nephropathy or hepatic cirrhosis), or mental illness were excluded. The cancer risk scoring system was based on the Harvard Risk Index [29], which was modified for the study according to risk factors specific to the Chinese population [30].

### 2.2. Sample Size Calculation

The minimum sample size estimation of DCEs is currently determined based on economics and effectiveness. According to Orme’s formula for the minimum sample size [31],
𝑁 > 500 c/(𝑡 × 𝑎),(1)
where 𝑐 is the product of the number of two attributes with the most levels, 𝑡 is the number of selection schemes, and 𝑎 is the number of selection items for each choice set; based on the experimental design of this study, 𝑐 is 9, 𝑡 is 8, 𝑎 is 3; therefore, the sample size of this study should be greater than 187.5. The sample size of similar studies conducted in Australia [13] and the Netherlands [32] ranged from 280 to 521, indicating that the relatively small sample size of this study has sufficient statistical power for a DCE.

### 2.3. Selection of Attributes and Levels

Attributes and levels of screening modalities were selected through a literature search and panel interviews. Broekhuizen et al. reported the relative weights and preferences of important attributes for lung cancer screening using a swing weighting questionnaire. The following attributes were included: sensitivity and specificity of screening modalities, radiation burden, duration of the screening procedure, time until screening results are communicated, mode of screening, and venue of screening programme [33]. Based on this, we selected seven attributes for the panel interview: sensitivity and specificity, screening tools, screening intervals, radiation burden, screening venues, and out-of-pocket expenses. The panel interview was organised with two health economics experts, two screening programme staff, and three respondents with a high risk of lung cancer. Sensitivity and specificity were excluded from the final questionnaire because these attributes seemed more important to screening designers but not the respondents and cognitively difficult for respondents with lower education levels to understand. Given current Chinese practice, the levels should show some degree of variation so that trade-offs are possible; the levels of each attribute were determined. The attributes finally included in the questionnaire were screening tools, screening intervals, radiation burden, screening venues, and out-of-pocket expenses. Details of the attributes and levels are given in Table 1.

### 2.4. Study Design

DCE is an attribute-based survey method for measuring benefits (utility), which could explore preference heterogeneity with complex and multi factors compared to single factor and multiple factors analysis. To put it simply, DCE is based on the assumption that an intervention can be described by its characteristics or attributes, which in turn are specified by several levels (for instance, sensitivity is a characteristic of a screening tool, and 80% sensitivity is one of its levels). Typically, a DCE consists of a series of choice sets where a respondent is asked to choose between two or more interventions that are defined by the same attributes but with varying levels. Patients’ choices provide information about the relative importance of these attributes and levels, through statistical modelling. A D-efficient experimental design with choice-sets was constructed using SAS v9.4 (SAS Institute, Cary, NC, USA) to help compile the questionnaire. The design consisted of 108 unique choice sets; D-efficiency was used as the primary criterion for identifying an optimal design. Unlike standard classical designs such as factorials and fractional factorials, D-optimal design matrices are usually not orthogonal and effect estimates are correlated; the optimality criterion in D-optimal design matrices results in minimising the generalised variance of the parameter estimates for a pre-specified model [34]. The final DCE of this study consisted of 10 choice sets with 1 of them overlapping to test the consistency of the responses. A written description of the attributes and levels was provided at the beginning of the DCE section, in addition to some related information about lung cancer screening; e.g., “Taking an extra blood test instead of taking LDCT alone could increase the accuracy of the screening results to avoid false-positive results”. Respondents were also asked about the level of self-perceived risk of lung cancer to explore preference variations between different subgroups (e.g., different level of self-perceived risk).

### 2.5. Survey Administration

The survey was carried out in Zhejiang Province, a highly developed province in Eastern China, at the forefront of innovation. We wanted the sample to represent a population at high risk of lung cancer so that the results are realistic. The screening programme staff and community volunteers for risk assessment enrolled all the respondents. Sample demographics and health status information were collected to estimate lung cancer risk. Only those assessed as having a high risk of lung cancer were asked to complete the DCE questionnaire, that is, to select (or opt out) between two unlabelled hypothetical strategies containing the selected attributes (Figure 1).

### 2.6. Statistical Analyses

The study analysis aimed to examine two main aims: 1) to identify the attributes that determine individuals’ preferences in lung cancer screening modalities, and 2) to explore the relationship between the variables and opt-out behaviour.

A mixed logit model was used to estimate public preferences between the two testing options. The mixed logit model extends the standard conditional logit model by allowing one or more of the parameters in the model to be randomly distributed to further explore heterogeneity. The usefulness of individual i of choosing alternative j in scenario t is
𝑈𝑖𝑗𝑡 = 𝐴𝑆𝐶 + 𝑥𝑖𝑡𝑗𝛽𝑖 + 𝜀𝑖𝑗𝑡 = 𝐴𝑆𝐶 + 𝑥𝑖𝑡𝑗(𝛽 + 𝜂𝑖) + 𝜀𝑖𝑗𝑡,(2)
where ASC is an alternative specific constant, 𝑥𝑖𝑡𝑗 is an observable attribute vector, 𝛽𝑖 is an individual-specific parameter vector, 𝛽 is the vector of mean attribute weights, and 𝜂 is an individual-specific deviation vector. Marginal willingness to pay (mWTP) was used to estimate the out-of-pocket expenses that an individual is willing to pay for the acquisition or improvement of a screening modality. In this study, the out-of-pocket cost attribute was regarded as a monetary attribute, and was included in the model as a continuous variable; thus, the respondent’s mWTP can be derived from the marginal substitution rate of non-monetary attributes and out-of-pocket costs according to the following formula [35]:(3)$${\mathrm{mWTP}}_{Xk}=\frac{\mathrm{MUXk}}{\mathrm{MUcost}},$$
where MUxk and MUcost were the marginal utility of attribute 𝑋𝑘 and out-of-pocket costs, respectively. Uptake rate prediction analysis, a very flexible post-evaluation tool, is a simple way to describe how the uptake probability changes with the change of attribute; it also provides a way to simulate interesting scenarios. The logic probability estimation equation of the individual choosing one scenario instead of another is as follows:(4)$$\mathrm{Pi}=\frac{e\beta 1\times 1i+\beta 2x2i+\cdots +\beta \mathrm{nxni}}{\sum e\beta 1x1j+\beta 2x2j+\cdots +\beta \mathrm{nxnj}}\;\forall i,j\in J,$$
where X𝑛𝑖 and X𝑛j were the attribute coefficient vectors of alternative i and alternative j, respectively. The heterogeneity of preferences in different subgroups (e.g., different level of self-perceived risk) was also explored.

Stepwise logistic regression was used to explore opt-out behaviour, the second aim. Stepwise multiple logistic regression analysis was performed to investigate the relationship between age, sex, marital status, the highest level of education, occupation, medical insurance, endowment insurance, smoking status, drinking status, self-perceived risk level and habit of getting health checks with opt-out behaviour. Statistical significance was set at *p* < 0·05. All analyses were conducted using STATA v14 (Stata Corp, College Station, TX, USA) [36].

## 3. Results

A total of 412 respondents completed all choice sets of the questionnaire, in which 393 (95.39%) provided the demographics for the analysis of opt-out behaviour and were therefore included in the data set. Of the respondents, 57.25% were male, and the mean age was 61.68 ± 6 years; 23.16% had no formal education, 58.02% had primary level education, and only 14.5% and 4.33% had attended junior middle school and high school, respectively. More than half of the respondents were farmers or fishermen. Half of the respondents claimed to have a health check habit, while the other half did not. Detailed information on sample demographics is presented in Table 2.

The results of the mixed logit analysis (without the opt-out choice) are shown in Table 3. The plus-minus sign of the coefficients indicates the direction of influence of each level from the attributes based on the choice of respondents. A positive coefficient indicates a positive preference, and a negative coefficient indicates a negative preference. The value of coefficient can be interpreted as preference weight, which represents the relative significance of respondents’ preference for each attribute level. On mixed logit analysis, the following variables were predictive of choice at 1% level of statistical significance: screening interval, screening venue, and out-of-pocket costs. The screening tool used and the level of radiation from the test did not appear to matter at the 5% level of statistical significance. The willingness to pay is defined as the amount the average respondent was willing to pay if they were to move from the base level of each attribute to every other level of that attribute [29]. For instance, the respondent may be willing to pay an extra RMB 154.01 to get a blood test in addition to an LDCT scan (Table 3).

Regarding the heterogeneity of different subgroups, the self-perceived risk level was assumed to be a significant driver of screening compliance; respondent preferences based on different levels of self-perceived risk are shown in Table 4. Among all high-risk respondents, only 6.10% perceived themselves as having an above-average risk of lung cancer. Most (74.35%) perceived themselves as having below average risk. Compared to the equal or above average risk subgroup, respondents who self-perceived as having below average risk considered the screening interval, screening venue, and out-of-pocket costs to be important (Table 4). 

The baseline for uptake rate analysis was set to screening modality by LDCT scan only, annual screening interval, use of mobile screening, low radiation, and low fees (around RMB 50). Based on the mixed logit regression results, the estimation equation was used to predict the possible change in the uptake rate under different screening modalities (Figure 2). The uptake rate would be increased by 0.40 compared with the baseline if the screening modality is low out-of-pocket expenses (RMB 50), LDCT + blood test, screening at a general hospital venue, or low radiation.

Opt-out was chosen as the most preferred of the three options in each choice set by 7.89% of all respondents, indicating an extremely low willingness to participate in a screening programme in any scenario. An examination of the demographic characteristics related to opt-out behaviour indicates that some of the characteristics appear to be statistically significant at the 1% or 5% level (Table 5). Those with no endowment insurance were more likely to opt out, while those who were older and housewives/househusbands, and those with a health check habit and with commercial endowment insurance were predicted to be less likely to opt out from a screening programme.

## 4. Discussion

We investigated the preferred modality for lung cancer screening in a Chinese high-risk population; this study was the first to use quantitative methods to examine preferences in lung cancer screening modalities in China. Our findings complement those from previous similar studies globally.

In summary, the key drivers of respondent choice of screening modality seemed to be the screening interval, screening venue and out-of-pocket costs. Wide standard deviation values indicate preference heterogeneity, particularly with regard to the type of screening tool used, the duration of the interval between the two screening tests, and the level of radiation from different screening modalities. However, the subgroup analysis of respondents with different levels of self-perceived risk indicates a slightly different pattern from the overall respondents. When respondents were divided into different subgroups by level of self-perceived risk, the preferred modality was either out-of-pocket costs or screening interval (respondents who self-perceived as having above average risk were less sensitive to these), and potentially, also the type of screening venue (respondents who self-perceived as having below average risk were more sensitive to this). In contrast, a survey conducted in Manchester, UK, reported that respondents were less likely to attend a similar hospital-based programme than mobile CT scanners [37]. A distrust of primary health service facilities may have contributed to this phenomenon in China, where the general hospital was the preferred screening venue in this study.

The estimation of willingness to pay demonstrated that respondents in general were willing to pay RMB 154.01– RMB 237.47 for an extra blood test in addition to a CT scan or for a screening test in a general hospital rather than in a mobile screening unit. As the nationwide lung cancer-screening programme has not been fully publicly funded, it is useful to have knowledge of this.

As regards the possible ways to increase the uptake rate, study results indicated that in general respondents could be motivated by using LDCT+ blood tests and the community or general hospital as the venue rather than the baseline setting. The preferred screening modality comprised screening by LDCT+ blood test once a year in a general hospital at a cost of RMB 50 for each screen. Based on our study findings, policymakers should focus on maintaining continuous screening tests for eligible populations, providing flexible screening tool combinations (e.g., LDCT and blood test) and using general hospitals as venues in order to increase the uptake of lung cancer screening. Bonuses or the reimbursement of medical insurance could also help reduce costs of screening, as the normal price of the LDCT and blood test is about RMB 750 in Zhejiang Province, a possible reason for the relatively low uptake rate.

While declining participation is obviously related to the personal characteristics of the respondents, the results of our stepwise regression analysis were consistent with the findings of similar studies, which suggest that age, occupation, self-perceived risk level, having a health check habit, and having an endowment insurance had a statistically significant impact on uptake. We therefore confirm previous reports that there might be some respondent subgroups that remain relatively hard to reach [29,38].

Of note, among all the respondents who had already been assessed as having a high risk of lung cancer, 67.25% continued to believe that they had below average risk of lung cancer. For reference, the incidence density varied between low-risk and high-risk populations, while the incidence density of the high-risk population was 2.37 higher than that of the low-risk population based on the Cancer Screening Program in Urban China (CanSPUC) from 2013 to 2019 in Zhejiang. A mistaken belief in low risk level might result in an underestimation of the need for lung cancer screening even among the eligible population or even the refusal to participate in a screening test. However, a cross-sectional survey of individuals from Tianjin Dagang Oilfield indicated that awareness of lung cancer risk factors was not low (77.10%) [39]. There is an underlying assumption that individuals do have an understanding of the risk factors related to lung cancer, but not enough knowledge of the risk assessment model. A literature search did not reveal any more publications related to the perceived risk of lung cancer in the Chinese population; this subject therefore deserves further investigation using, in particular, qualitative research methods.

## 5. Conclusions

Our study focused on the design and infrastructure of a lung cancer-screening programme in a Chinese setting. We observed that the screening interval was a determining factor in the uptake in both respondents in general and in subgroups, which poses a considerable challenge to the implementation of a sustained regular screening programme. The preferred screening modality comprised screening by LDCT + blood test once a year in a general hospital at a cost of RMB 50 for each screen; this is not completely consistent with current Chinese practice. There is considerable variance between real risk and self-perceived risk of lung cancer among respondents; further research on the attitudes towards and awareness of lung cancer risk assessment among Chinese populations is therefore required. The results provide insight into the decision-making process regarding the decision of what kind of lung cancer screening programme to take. This information can be used to help policymakers further design the screening modality by selecting the attribute levels that maximize screening attendance and provide more information about the screening invitees who are hard to reach.

## Figures and Tables

**Figure 1 cancers-13-06110-f001:**
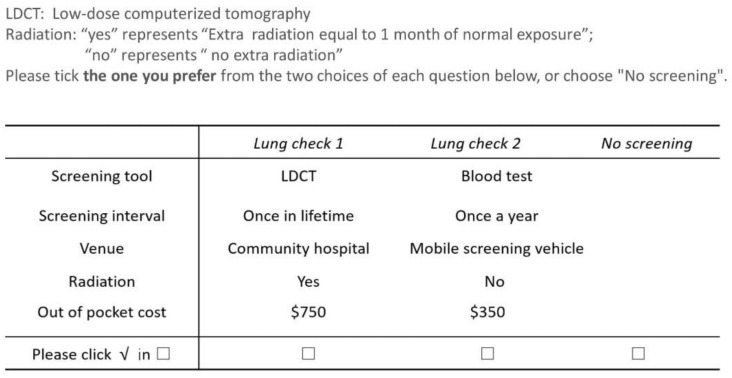
An example choice set.

**Figure 2 cancers-13-06110-f002:**
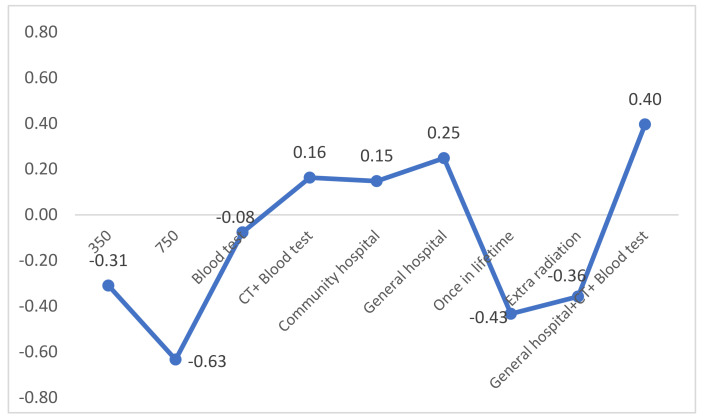
Uptake rate under different screening modalities.

**Table 1 cancers-13-06110-t001:** Included attributes and levels.

Attributes	Levels	Descriptions
Screening tools	tool0	LDCT
	tool1	Blood test
	tool2	LDCT + Blood test
Screening intervals	interval0	Once a year
	interval1	Once in lifetime
Venues	Venue0	Mobile screening vehicle
	Venue1	Community hospital
	Venue2	General hospital
Radiation	radiation0	No extra radiation
	radiation1	Extra radiation equal to 1 month
Out-of-pocket cost	otp0	$50
	otp1	$350
	otp2	$750

**Table 2 cancers-13-06110-t002:** Sample demographics.

Characteristic	Level	Number (or Mean) (% Unless Stated)
Gender	Male	225 (57.25)
	Female	168 (42.75)
Age	Current	61.68 (SD:6) ^1^
BMI	<18.5	19 (4.83)
	18.5–23	172 (43.77)
	>23	202 (51.40)
Highest level of education	No formal education	91 (23.16)
	Primary school	228 (58.02)
	Junior middle school	57 (14.50)
	High school	17 (4.33)
Marital status	Married	383 (97.46)
	Single and others	10 (2.54)
Occupation	Enterprise personnel	7 (1.78)
	Farmer/fisherman	266 (67.68)
	Worker or production personnel	30 (7.63)
	Housework	90 (22.90)
Smoking status	Non-smoker	196 (49.87)
	Smoker	154 (39.19)
	Former smoker	43 (10.94)
Drinking status	Non-drinker	335 (85.24)
	Drinker	58 (14.76)
Family history of any cancer	No	276 (70.23)
	Yes	117 (29.77)
Medical insurance	Basic Medical Insurance System for Urban Residents	53 (13.49)
	New Rural Cooperative Medical Insurance	305 (77.61)
	No medical insurance	31 (7.89)
	Medical Insurance System for Urban Employees	4 (1.02)
Endowment insurance	Basic Endowment Insurance for Urban Employees	7 (1.80)
	Endowment insurance for flexible employees	5 (1.27)
	Social Endowment Insurance for Urban and Rural Residents	18 (4.58)
	New Rural Society Endowment Insurance	77 (19.59)
	Commercial Endowment Insurance	4 (1.02)
	Other Endowment Insurance	11 (2.80)
	No Endowment Insurance	271 (68.96)
Self-perceived risk Level of lung cancer	Below average	257 (65.39)
	Equal to the average	104 (26.46)
	Above average	32 (8.14)
Cancer patients in acquaintance	Yes	34 (8.65)
	No	359 (91.35)
Habit of health check	Yes	164 (41.73)
	No	229 (58.27)

^1^ Note: SD indicates standard deviation.

**Table 3 cancers-13-06110-t003:** Results of mixed logit analysis (all respondents).

Variable	Level	Coefficient (SE)	Standard Deviation (SE)	Willingness to pay (RMB, 95%CI)
Screening tool (base is CT)	Blood test	−0.1539(0.2915)	1.3290(0.2625) *	−72.078(−325.164,213.775)
	CT+ blood test	0.3290(0.2817)	2.3455(0.2824) *	154.010(−120.210,431.630)
Screening interval (base is once a year	Once in lifetime	−0.9284(0.1476) *	0.4856(0.1821) *	−434.789(−620.196, −293.836)
Venue (base is mobile screening vehicle)	Community hospital	0.2974(0.1475) †	−0.2350(0.2442)	139.291(5.0674,277.3748)
	General hospital	0.5071(0.1360) *	−0.0457(0.2094)	237.470(124.9257,363.4297)
Radiation (base is no extra radiation)	Extra radiation equal to 1 month of normal exposure	−0.7481(0.7182)	1.1780(0.3718) *	−350.343(−1082.626,321.302)
Out-of-pocket cost	Continuous	−0.0021(0.0003) *	-	-
Log likelihood		−716.4839	-	-

Note: Statistical significance is noted at the 1% level (*) and the 5% level (†). CT indicates computerized tomography; SE, standard error.

**Table 4 cancers-13-06110-t004:** Results of mixed logit analysis (different self-perceived risk level).

Variables and Levels	Self-Perceived Risk below Average		Self-Perceived Risk Equal to Average		Self-Perceived Risk above Average	
	Coefficient (SE)	Standard Deviation (SE)	Coefficient (SE)	Standard Deviation (SE)	Coefficient (SE)	Standard Deviation (SE)
Out-of-pocket cost	−0.0020 (0.0004) *	-	−0.0038 (0.0011) *	-	−0.0034 (0.0022)	-
Screening tools: LDCT only (base)						
Blood test	−0.4087 (0.3519)	1.6106 (0.3186) *	0.6956 (0.7893)	0.1502 (0.6906)	0.2073 (1.4668)	−0.4674 (1.2589)
CT+ blood test	0.4446 (0.3165)	2.0038 (0.3120) *	−0.2063 (1.0518)	3.5542 (0.9633) *	4.2413 (2.3696)	4.3340 (1.8522) †
Screening interval: once a year (base)						
Once in a lifetime	−1.0709 (0.1955) *	0.4699 (0.2206) †	−0.7470 (0.4115)	1.1755 (0.5899) †	−6.5654 (3.4343)	3.6777 (2.0027)
Venue: mobile screening vehicle (base)						
Community hospital	0.3533 (0.1710)	−0.0177 (0.3051)	0.2143 (0.4529)	−0.9283 (0.4516) †	−2.0101 (1.4104)	−0.8934 (0.5756)
General hospital	0.5701 (0.1596) *	−0.2760 (0.2780)	0.2663 (0.3603)	−0.1411 (0.4620)	0.5249 (0.8210)	0.0468 (1.4744)
Radiation: no extra radiation (base)						
Extra radiation equal to 1 month of normal exposure	−1.9241 (2.2755)	2.0091 (1.4887)	0.1583 (0.9634)	0.9617 (0.3823) †	0.2549 (1.7344)	0.7460 (0.8204)
Log likelihood	−523.5244		−135.5215		−35.7729	
Number of observations	2000		526		164	

Note: Statistical significance is noted at the 1% level (*) and the 5% level (†).

**Table 5 cancers-13-06110-t005:** Predicting opt-out behaviour.

Variable	Level	Coefficients (SE)
Age	Continuous	−0.0049 (0.0021) †
Highest level of education	High school	−0.1083 (0.0652)
Self-perceived risk Level	Equal to the average	−0.1116 (0.0294) *
Occupation	Farmers	−0.0801 (0.0463)
	Housework	−0.1313 (0.0510) †
Habit of health check	Yes	−0.0595 (0.0270) †
Medical insurance	Medical Insurance System for Urban Employees	−0.0827 (0.0593)
	New Rural Cooperative Medical Insurance	0.0593 (0.0373)
Endowment insurance	Commercial Endowment Insurance	−0.2992 (0.1301) †
	No Endowment Insurance	0.0730 (0.0301) †

Note: Statistical significance is noted at the 1% level (*) and the 5% level (†).

## Data Availability

The data presented in this study are available on request from the corresponding author.

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
