# Peer review of "Preferred Lung Cancer Screening Modalities in China: A Discrete Choice Experiment"

_cancers, 2021, doi:10.3390/cancers13236110_

Round 1

Reviewer 1 Report

This study aims to investigate factors that may improve adherence to lung cancer screening in China among patients aged 50-74 years at high risk for lung cancer. A total of 412 people responded, of which 393 were usable.
The modalities that seemed to have the greatest impact on adherence to screening were: interval screening; screening venue; out-of-pocket costs.
It is important to note that of this cohort of high-risk patients, approximately 67% considered themselves to be at low or moderate risk.
The most consensual modality was a screening by LDCT with blood test once a year, in a general hospital with a cost of RMB50 at each screening, which does not seem to be feasible with the current practices in China according to the authors.

This study finds common results with those conducted in other countries.
This study will help guide the screening policy for China and try to find a consensus to make screening attractive and as inexpensive as possible for society.
Nevertheless, we can wonder whether responders are aware of the possible results of the blood tests and the therapeutic implication afterwards. Moreover, the blood test in question is not detailed, and we do not know what they will look for (notably at the mutational level) to demonstrate localized lung cancer.

Author Response

Thank you for your comments.

First, before the screening program enrolment, all the potential participants were gathered to a brief lecture of lung cancer screening. We informed the respondents that, “Taking an extra blood test instead of taking LDCT alone could increase the accuracy of the screening results to avoid false-positive results.” and if lung cancer was detected, what the subsequent steps would be. Specific figures about the sensitivity and specificity difference between the two screening tools were not provided to respondents in the interest of comprehensibility. The pilot survey indicated that it might be cognitively difficult for respondents with lower levels of education to understand the specificities, and we wished to make the questionnaire easy to understand.

Second, the “blood test” in our study was the EarlyCDT®-Lung Test, an immunobiomarker test that aids early detection of lung cancer. In addition, to increase readability and avoid confusion, we decided not to display the specific production name in the whole survey. We have now added details of the information provided to the participants on the blood tests in the study design section. The changes made are as follows:

Page 4, Lines 152-155:

A written description of the attributes and levels was provided at the beginning of the DCE section, in addition to some related information about lung cancer screening; e.g., “Taking an extra blood test instead of taking LDCT alone could increase the accuracy of the screening results to avoid false-positive results”.

Reviewer 2 Report

Recently I was requested to review a paper entitled “Preferred lung cancer screening modalities in China: A discrete choice experiment”. Lung cancer screening is an important issue in the secondary prevention of this deadly disease. China is an important country on the map of lung cancer mortality. Studies on LDCT screening in this country are required and have high citation potential. The concept of DCE is relatively new. I find it interesting in designing screening protocols the most suitable for general population. The paper is well written and easy to understand. I do not have many significant remarks concerning the style.

I have only one general remark. We shall remember that the main audience of this study would be clinicians and scientists dealing with lung cancer screening. The concept of DCE maybe not be commonly recognized by most of them. I would like to ask the authors to consider a brief presentation of the methodology of this instrument. Scientific soundness would be increased if the interpretation of results presented in the tables would be explained more thoroughly in the text i.e. Table 3 “Coefficient, SD, Willingness to pay”. It may be difficult to interpret that for some of the clinicians who would be attracted by the paper anyway.

Nevertheless, I would like to congratulate the authors on the manuscript and I await its publication.

Author Response

Thank you for your comments.

We completely agree with this perspective. We have now added a brief introduction to DCE in the Study Design section to help the clinicians better understand this method. Additionally, a detailed interpretation has been added for Table 3. Overall, we have also improved on the English throughout the manuscript. The changes made are as follows:

Page 4, Line 136-143

To put it simply, DCE is based on the assumption that an intervention can be described by its characteristics or attributes, which in turn are specified by several levels (for instance, sensitivity is a characteristic of a screening tool, and 80% sensitivity is one of its levels). Typically, a DCE consists of a series of choice sets where a respondent is asked to choose between two or more interventions that are defined by the same attributes but with varying levels. Patients’ choices provide information about the relative importance of these attributes and levels, through statistical modelling.

Page 8, Line 217-221

The plus-minus sign of the coefficients indicates the direction of influence of each level from the attributes based on the choice of respondents. A positive coefficient indicates a positive preference, and a negative coefficient indicates a negative preference. The value of coefficient can be interpreted as preference weight, which represents the relative significance of respondents' preference for each attribute level.
